# Adversarial Training Generalizes Data-dependent Spectral Norm Regularization

## Abstract

We establish a theoretical link between adversarial training and operator norm regularization for deep neural networks. Specifically, we present a data-dependent variant of spectral norm regularization and prove that it is equivalent to adversarial training based on a specific $\ell_2$-norm constrained projected gradient ascent attack. This fundamental connection confirms the long-standing argument that a network's sensitivity to adversarial examples is tied to its spectral properties and hints at novel ways to robustify and defend against adversarial attacks. We provide extensive empirical evidence to support our theoretical results.

## 1 Introduction

Deep neural networks have been used with great success for perceptual tasks such as image classification (Simonyan & Zisserman, 2014; LeCun et al., 2015) or speech recognition (Hinton et al., 2012). While they are known to be robust to random noise, it has been shown that the accuracy of deep nets dramatically deteriorates in the face of so-called adversarial examples (Biggio et al., 2013; Szegedy et al., 2013; Goodfellow et al., 2014), i.e. small perturbations of the input signal, often imperceptible to humans, that are sufficient to induce large changes in the model output. This apparent vulnerability is worrisome as deep nets start to proliferate in the real-world, including in safety-critical deployments. Consequently, there has been a surge in methods that find adversarial perturbations (Sabour et al., 2015; Papernot et al., 2016; Kurakin et al., 2016; Moosavi Dezfooli et al., 2016; Moosavi-Dezfooli et al., 2017; Madry et al., 2017; Athalye et al., 2018).

The most direct strategy of robustification, called adversarial training, aims to harden a machine learning model by immunizing it against an adversary that maliciously corrupts training examples before passing them to the model (Goodfellow et al., 2014; Kurakin et al., 2016; Miyato et al., 2015; 2017; Madry et al., 2017). A different strategy of defense is to detect whether the input has been perturbed by detecting characteristic regularities either in the adversarial perturbations themselves or in the network activations they induce (Grosse et al., 2017; Feinman et al., 2017; Xu et al., 2017; Metzen et al., 2017; Carlini & Wagner, 2017; Roth et al., 2019).

Despite practical advances in finding adversarial examples and defending against them, it is still an open question whether (i) adversarial examples are unavoidable, i.e. no robust model exists, cf. (Fawzi et al., 2018; Gilmer et al., 2018), (ii) learning a robust model requires too much training data, cf. (Schmidt et al., 2018), (iii) learning a robust model from limited training data is possible but computationally intractable (Bubeck et al., 2018), or (iv) we just have not found the right training algorithm yet, i.e. adversarial examples exist because of intrinsic flaws of the model or learning objective that can ultimately be overcome.

In this work, we investigate the origin of adversarial vulnerability in neural networks by focusing on the attack algorithms used to find adversarial examples. In particular, we make the following contributions:

- We present a data-dependent variant of spectral norm regularization that directly regularizes large singular values of a neural network in regions that are supported by the data, as opposed to existing methods that regularize a global, data-independent upper bound.

- We establish a theoretical link between adversarial training and operator norm regularization for deep neural networks. Specifically, we prove that data-dependent spectral norm regularization is equivalent to adversarial training based on a specific $\ell_2$-norm constrained projected gradient ascent attack.

- We conduct extensive empirical evaluations showing that (i) adversarial perturbations align with dominant singular vectors, (ii) adversarial training and data-dependent spectral norm regularization dampen the singular values, and (iii) both training methods give rise to models that are significantly more linear around data points than normally trained ones.

## 2 RELATED WORK

The idea that a conservative measure of the sensitivity of a network against adversarial examples can be obtained by computing the spectral norm of the individual weight layers appeared already in the seminal work of Szegedy et al. (2013). A number of works have since suggested to regularize the spectral norm (Yoshida & Miyato, 2017; Miyato et al., 2018; Bartlett et al., 2017; Farnia et al., 2018) and Lipschitz constant (Cisse et al., 2017; Hein & Andriushchenko, 2017; Tsuzuku et al., 2018; Raghunathan et al., 2018) as a means to improve model robustness against adversarial attacks. In the same vein, training methods based on input gradient regularization have been proposed (Gu & Rigazio, 2014; Lyu et al., 2015; Cisse et al., 2017).

The most direct and popular strategy of robustification, however, is to use adversarial examples as data augmentation during training (Goodfellow et al., 2014; Shaham et al., 2015; Kurakin et al., 2016; Miyato et al., 2017; Madry et al., 2017). Adversarial training can be viewed as a variant of (distributionally) robust optimization (El Ghaoui & Lebret, 1997; Xu et al., 2009; Bertsimas & Copenhaver, 2018; Namkoong & Duchi, 2017; Sinha et al., 2017; Gao & Kleywegt, 2016) where a machine learning model is trained to minimize the worst-case loss against an adversary that can shift the entire training data within an uncertainty set. Interestingly, for certain problems and uncertainty sets, such as for linear regression and induced matrix norm balls, robust optimization has been shown to be equivalent to regularization (El Ghaoui & Lebret, 1997; Xu et al., 2009; Bertsimas & Copenhaver, 2018; Bietti et al., 2018). Similar results on the equivalence of robustness and regularization have been obtained also for (kernelized) SVMs (Xu et al., 2009).

More recently, related works have started to develop a learning theory for robust optimization, including Lipschitz-sensitive generalization bounds (Neyshabur et al., 2015) and spectrally-normalized margin bounds for neural networks (Bartlett et al., 2017), particularly as bounds on the spectral norm or Lipschitz constant can easily be translated to bounds on the minimal perturbation required to fool a machine learning model.

We extend these lines of work by establishing a theoretical link between adversarial training and data-dependent spectral norm regularization. This fundamental connection confirms the long-standing argument that a network's sensitivity to adversarial examples is tied to its spectral properties and opens the door for adversarially robust generalization bounds via spectral norm based ones.

## 3 BACKGROUND

### 3.1 GLOBAL SPECTRAL NORM REGULARIZATION

In this section we rederive spectral norm regularization à la Yoshida & Miyato (2017), while also setting up the notation for later. Let $\mathbf{x}$ and $y$ denote input-label pairs generated from a data distribution $P$. Let $f : \mathcal{X} \subset \mathbb{R}^n \to \mathbb{R}^d$ denote the logits of a $\theta$-parameterized piecewise linear classifier, i.e. $f(\cdot) = \mathbf{W}^L \phi^{L-1}(\mathbf{W}^{L-1} \phi^{L-2}(\dots) + \mathbf{b}^{L-1}) + \mathbf{b}^L$, where $\phi^\ell$ is the activation function, and $\mathbf{W}^\ell$, $\mathbf{b}^\ell$ denote the layer-wise weight matrix[1] and bias vector, collectively denoted by $\theta$. Let us furthermore assume that each activation function is a ReLU (the argument can easily be generalized to other piecewise linear activations). In this case, the activations $\phi^\ell$ act as input-dependent diagonal matrices $\Phi_{\mathbf{x}}^\ell := \text{diag}(\phi_{\mathbf{x}}^\ell)$, where an element in the diagonal $\phi_{\mathbf{x}}^\ell := \mathbf{1}(\tilde{\mathbf{x}}^\ell \geq 0)$ is one if the corresponding pre-activation $\tilde{\mathbf{x}}^\ell := \mathbf{W}^\ell \phi^{\ell-1}(\cdot) + \mathbf{b}^\ell$ is positive and equal to zero otherwise.

---

[1]Note that convolutional layers can be constructed as matrix multiplications by converting the convolution operator into a Toeplitz matrix.

Following Raghu et al. (2017), we call $\phi_{\mathbf{x}} := (\phi_{\mathbf{x}}^1, \ldots, \phi_{\mathbf{x}}^{L-1}) \in \{0,1\}^m$ the "activation pattern", where $m$ is the number of neurons in the network. For any activation pattern $\phi \in \{0,1\}^m$ we can define the preimage $X(\phi) := \{\mathbf{x} \in \mathbb{R}^n : \phi_{\mathbf{x}} = \phi\}$, inducing a partitioning of the input space via $\mathbb{R}^n = \bigcup_\phi X(\phi)$. Note that some $X(\phi) = \emptyset$, as not all combinations of activiations may be feasible. See Figure 1 in (Raghu et al., 2017) or Figure 3 in (Novak et al., 2018) for an illustration of ReLU tesselations of the input space.

We can linearize $f$ within a neighborhood around $\mathbf{x}$ as follows

$$f(\mathbf{x} + \Delta\mathbf{x}) \simeq f(\mathbf{x}) + \mathbf{J}_{f(\mathbf{x})}\Delta\mathbf{x} \,, \quad (\text{with equality if } \mathbf{x} + \Delta\mathbf{x} \in X(\phi_{\mathbf{x}})) \,, \tag{1}$$

where $\mathbf{J}_{f(\mathbf{x})}$ denotes the Jacobian of $f$ at $\mathbf{x}$

$$\mathbf{J}_{f(\mathbf{x})} = \mathbf{W}^L \cdot \Phi_{\mathbf{x}}^{L-1} \cdot \mathbf{W}^{L-1} \cdot \Phi_{\mathbf{x}}^{L-2} \cdots \Phi_{\mathbf{x}}^1 \cdot \mathbf{W}^1 \,. \tag{2}$$

We have the following bound for $||\Delta\mathbf{x}||_2 \neq 0$

$$\frac{||f(\mathbf{x} + \Delta\mathbf{x}) - f(\mathbf{x})||_2}{||\Delta\mathbf{x}||_2} \simeq \frac{||\mathbf{J}_{f(\mathbf{x})}\Delta\mathbf{x}||_2}{||\Delta\mathbf{x}||_2} \leq \sigma(\mathbf{J}_{f(\mathbf{x})}) := \sup_{||\Delta\mathbf{x}||_2 \neq 0} \frac{||\mathbf{J}_{f(x)}\Delta\mathbf{x}||_2}{||\Delta\mathbf{x}||_2} \,, \tag{3}$$

where $\sigma(\mathbf{J}_{f(\mathbf{x})})$ is the *spectral norm* (largest singular value) of the linear operator $\mathbf{J}_{f(\mathbf{x})}$. From a robustness perspective we want $\sigma(\mathbf{J}_{f(\mathbf{x})})$ to be small in regions that are supported by the data.

Based on the decomposition in Equation 2 and the non-expansiveness of the activations, $\sigma(\Phi_{\mathbf{x}}^\ell) \leq 1$ for every $\ell \in \{1, ..., L-1\}$, Yoshida & Miyato (2017) suggest to upper-bound the spectral norm of the Jacobian by the product of the spectral norms of the individual weight matrices

$$\sigma(\mathbf{J}_{f(\mathbf{x})}) \leq \prod_{\ell=1}^L \sigma(\mathbf{W}^\ell) \quad , \forall \mathbf{x} \in \mathcal{X} \,. \tag{4}$$

The layer-wise spectral norms $\sigma^\ell := \sigma(\mathbf{W}^\ell)$ can be computed iteratively using the power method[3]. Starting with a random vector $\mathbf{v}_0$, the power method iteratively computes

$$\mathbf{u}_k^\ell \leftarrow \tilde{\mathbf{u}}_k^\ell/||\tilde{\mathbf{u}}_k^\ell||_2 \,, \; \tilde{\mathbf{u}}_k^\ell \leftarrow \mathbf{W}^\ell \mathbf{v}_{k-1}^\ell \,, \quad \mathbf{v}_k^\ell \leftarrow \tilde{\mathbf{v}}_k^\ell/||\tilde{\mathbf{v}}_k^\ell||_2 \,, \; \tilde{\mathbf{v}}_k^\ell \leftarrow (\mathbf{W}^\ell)^\top \mathbf{u}_k^\ell \,. \tag{5}$$

The (final) singular value can be obtained via $\sigma_k^\ell = (\mathbf{u}_k^\ell)^\top \mathbf{W}^\ell \mathbf{v}_k^\ell$.

Yoshida & Miyato (2017) suggest to turn this upper-bound into a global (data-independent) regularizer by learning the parameters $\theta$ via the following penalized empirical risk minimization

$$\min \theta \rightarrow \mathbf{E}_{(\mathbf{x},y)\sim\hat{P}}\left[\ell(y, f(\mathbf{x}))\right] + \frac{\lambda}{2}\sum_{\ell=1}^L \sigma(\mathbf{W}^\ell)^2 \,, \tag{6}$$

where $\ell(\cdot, \cdot)$ denotes an arbitrary classification loss. Note, since the parameter gradient of $\sigma(\mathbf{W}^\ell)^2/2$ is $\sigma^\ell \mathbf{u}^\ell(\mathbf{v}^\ell)^\top$, with $\sigma^\ell$, $\mathbf{u}^\ell$ and $\mathbf{v}^\ell$ being the dominant singular value and singular vectors of $\mathbf{W}^\ell$ (approximated via the power method), Yoshida & Miyato (2017) global spectral norm regularizer effectively adds a term $\lambda\sigma^\ell \mathbf{u}^\ell(\mathbf{v}^\ell)^\top$ for each layer $\ell \in \{1, ..., L\}$ to the parameter gradient of the loss function. In terms of computational complexity, because the global regularizer decouples from the empirical loss, the power-method iterations can be amortized across data-points and a single power method iteration per parameter update step usually suffices in practice (Yoshida & Miyato, 2017).

## 3.2 Global vs. Local Regularization

The advantage of global bounds is that they trivially generalize from the training to the test set. The problem however is that they can be arbitrarily loose, e.g. penalizing the spectral norm over irrelevant regions of the ambient space. To illustrate this, consider the ideal robust classifier that is essentially piecewise constant on class-conditional regions, with sharp transitions between the classes. The global spectral norm will be heavily influenced by the sharp transition zones, whereas a local data-dependent bound can adapt to regions where the classifier is approximately constant (Hein & Andriushchenko, 2017). We would therefore expect a global regularizer to have the largest effect in the empty parts of the input space. A local regularizer, on the contrary, has its main effect around the data manifold.

# 4 ADVERSARIAL TRAINING GENERALIZES SPECTRAL NORM REGULARIZATION

## 4.1 DATA-DEPENDENT SPECTRAL NORM REGULARIZATION

We now show how to directly regularize the data-dependent spectral norm of the Jacobian $\mathbf{J}_{f(\mathbf{x})}$. Under the assumption that the dominant singular value is non-degenerate[2], the problem of computing the largest singular value and the corresponding left and right singular vectors can efficiently be solved via the power method. Let $\mathbf{v}_0$ be a random vector or an approximation to the dominant right singular vector of $\mathbf{J}_{f(\mathbf{x})}$. The power method iteratively computes

$$\begin{aligned}
\mathbf{u}_k \leftarrow \tilde{\mathbf{u}}_k/||\tilde{\mathbf{u}}_k||_2\,, \quad \tilde{\mathbf{u}}_k \leftarrow \mathbf{J}_{f(\mathbf{x})}\mathbf{v}_{k-1} = \mathbf{W}^L \cdot \Phi_{\mathbf{x}}^{L-1} \cdots \Phi_{\mathbf{x}}^1 \cdot \mathbf{W}^1\,\mathbf{v}_{k-1} \quad \text{(forward pass)} \\
\mathbf{v}_k \leftarrow \tilde{\mathbf{v}}_k/||\tilde{\mathbf{v}}_k||_2\,, \quad \tilde{\mathbf{v}}_k \leftarrow \mathbf{J}_{f(\mathbf{x})}^{\top}\mathbf{u}_k = \nabla_{\mathbf{x}}(f(\mathbf{x})^{\top}\mathbf{u}_k) \qquad\qquad\quad \text{(backward pass)}
\end{aligned} \tag{7}$$

The (final) singular value can be computed via $\sigma_k = \mathbf{u}_k^{\top}\mathbf{J}_{f(\mathbf{x})}\mathbf{v}_k$. Note that the right singular vector $\mathbf{v}_k$ gives the direction in input space that corresponds to the steepest ascent of $f(\mathbf{x})$ along $\mathbf{u}_k$.

We can turn this into a regularizer by learning the parameters $\theta$ via the following Jacobian-based spectral norm penalized empirical risk minimization

$$\min \theta \rightarrow \mathbf{E}_{(\mathbf{x},y)\sim\hat{P}}\left[\ell(y,f(\mathbf{x})) + \frac{\tilde{\lambda}}{2}(\mathbf{u}^{\top}\mathbf{J}_{f(\mathbf{x})}\mathbf{v})^2\right]\,, \tag{8}$$

where $\mathbf{u}$ and $\mathbf{v}$ are the data-dependent singular vectors of $\mathbf{J}_{f(\mathbf{x})}$, computed via Equation 7.

By optimality / stationarity[3] $(\mathbf{u}^{\top}\mathbf{J}_{f(\mathbf{x})}\mathbf{v})^2 = \mathbf{v}^{\top}\mathbf{J}_{f(\mathbf{x})}^{\top}\mathbf{J}_{f(\mathbf{x})}\mathbf{v} = ||\mathbf{J}_{f(\mathbf{x})}\mathbf{v}||_2^2$ and linearization $\epsilon\mathbf{J}_{f(\mathbf{x})}\mathbf{v} \simeq f(\mathbf{x} + \epsilon\mathbf{v}) - f(\mathbf{x})$ (which holds with equality if $\mathbf{x} + \epsilon\mathbf{v} \in X(\phi_{\mathbf{x}})$), we can regularize learning also via the following sum-of-squares based spectral norm regularizer

$$\min \theta \rightarrow \mathbf{E}_{(\mathbf{x},y)\sim\hat{P}}\left[\ell(y,f(\mathbf{x})) + \frac{\lambda}{2}||f(\mathbf{x} + \epsilon\mathbf{v}) - f(\mathbf{x})||_2^2\right]\,, \tag{9}$$

where the data-dependent singular vector $\mathbf{v}$ of $\mathbf{J}_{f(\mathbf{x})}$ is computed via Equation 7, and $\tilde{\lambda} = \lambda\epsilon^2$. Both variants can readily be implemented in modern deep learning frameworks. We found the sum-of-squares based spectral norm regularizer to be more numerically stable than the Jacobian based one, which is why we used this variant in our experiments.

In terms of computational complexity, the data-dependent regularizer is equally expensive as PGA-based adversarial training, and both are a constant (number of power method iterations) times more expensive than the data-independent variant, plus an overhead that depends on the batch size, which is usually mitigated in modern frameworks by parallelizing computations across a batch of data.

## 4.2 POWER METHOD FORMULATION OF ADVERSARIAL TRAINING

Adversarial training (Goodfellow et al., 2014; Kurakin et al., 2016; Madry et al., 2017) aims to improve the robustness of a machine learning model by training it against an adversary that independently perturbs each training example subject to a proximity constraint, e.g. in $\ell_p$-norm,

$$\min \theta \rightarrow \mathbf{E}_{(\mathbf{x},y)\sim\hat{P}}\left[\ell(y,f(\mathbf{x})) + \lambda \max_{\mathbf{x}^*\in\mathcal{B}_\epsilon^p(\mathbf{x})} \ell_{\text{adv}}(y,f(\mathbf{x}^*))\right]\,. \tag{10}$$

where $\ell_{\text{adv}}(\cdot,\cdot)$ denotes the loss function used to find adversarial perturbations (does not need to be the same as the classification loss $\ell(\cdot,\cdot)$).

The adversarial example $\mathbf{x}^*$ is typically computed iteratively, e.g. via $\ell_2$-norm constrained projected gradient ascent (Madry et al., 2017; Kurakin et al., 2016) (the general $\ell_p$-norm constrained case is similar)

$$\mathbf{x}_k = \Pi_{\mathcal{B}_\epsilon^2(\mathbf{x})}\left(\mathbf{x}_{k-1} + \alpha\frac{\nabla_{\mathbf{x}}\ell_{\text{adv}}(y,f(\mathbf{x}_{k-1}))}{||\nabla_{\mathbf{x}}\ell_{\text{adv}}(y,f(\mathbf{x}_{k-1}))||_2}\right)\,, \quad \mathbf{x}_0 \sim \mathcal{U}(\mathcal{B}_\epsilon^2(\mathbf{x}))\,, \tag{11}$$

---

[2]Due to numerical errors, we can safely assume that the dominant singular value is non-degenerate.
[3]$\mathbf{u} = \mathbf{J}_{f(\mathbf{x})}\mathbf{v}/||\mathbf{J}_{f(\mathbf{x})}\mathbf{v}||_2$

where $\Pi_{\mathcal{B}_\epsilon^2(\mathbf{x})}$ is the projection operator into the norm ball $\mathcal{B}_\epsilon^2(\mathbf{x}) := \{\mathbf{x}^* : ||\mathbf{x}^* - \mathbf{x}||_2 \leq \epsilon\}$, $\alpha$ is a step-size or weighting factor, trading off the previous iterate $\mathbf{x}_{k-1}$ with the current gradient direction $\nabla_\mathbf{x} \ell_{\mathrm{adv}}(y, f(\mathbf{x}_{k-1}))/||\nabla_\mathbf{x}\ell_{\mathrm{adv}}(y, f(\mathbf{x}_{k-1}))||_2 =: \mathbf{v}_k$, and $y$ is the true or predicted label. For targeted attacks the sign in front of $\alpha$ is flipped, so as to descend the loss function into the direction of the target label.

By the chain-rule, the computation of the gradient-step $\mathbf{v}_k$ can be decomposed into a logit-gradient and a Jacobian vector product, while the projection into the $\ell_2$-norm ball $\Pi_{\mathcal{B}_\epsilon^2(\mathbf{x})}$ can be expressed as a normalization (see Section A.2 in the Appendix). The $\ell_2$-norm constrained projected gradient ascent attack can thus equivalently be written in the following power method like form (the normalization of $\tilde{\mathbf{u}}_k$ is optional and can be absorbed into the normalization of $\tilde{\mathbf{v}}_k$)

$$\mathbf{u}_k \leftarrow \tilde{\mathbf{u}}_k/||\tilde{\mathbf{u}}_k||_2 , \quad \tilde{\mathbf{u}}_k \leftarrow \nabla_\mathbf{z} \ell_{\mathrm{adv}}(y, \mathbf{z})|_{\mathbf{z}=f(\mathbf{x}_{k-1})} \qquad \text{(forward pass)}$$

$$\mathbf{v}_k \leftarrow \tilde{\mathbf{v}}_k/||\tilde{\mathbf{v}}_k||_2 , \quad \tilde{\mathbf{v}}_k \leftarrow \mathbf{J}_{f(\mathbf{x}_{k-1})}^\top \mathbf{u}_k = \nabla_\mathbf{x}(f(\mathbf{x}_{k-1})^\top \mathbf{u}_k) \qquad \text{(backward pass)} \qquad (12)$$

$$\mathbf{x}_k \leftarrow \Pi_{\mathcal{B}_\epsilon^2(\mathbf{x})}(\mathbf{x}_{k-1} + \alpha\mathbf{v}_k) \qquad \text{(projection)}$$

Note that the logit-gradient $\nabla_\mathbf{z}\ell_{\mathrm{adv}}(y, \mathbf{z})|_{\mathbf{z}=f(\mathbf{x}_{k-1})}$ can be computed in a single forward pass, by directly expressing it in terms of the arguments of the adversarial loss.

Comparing the update equations for projected gradient ascent based adversarial training with those of data-dependent spectral norm regularization, we can see that *adversarial training generalizes spectral norm regularization in two ways: (i) via the choice of the adversarial loss function and (ii) by iterating* $\mathbf{x}_k$ *within the norm ball* (which also introduces an additional parameter $\alpha$).

The adversarial loss function determines the direction $\mathbf{u}_k$ of the directional derivative $\nabla_\mathbf{x}(f(\mathbf{x}_{k-1})^\top \mathbf{u}_k)$, cf. Section A.3 in the Appendix for an example using the softmax cross-entropy loss. The following theorem shows that adversarial training based on a specific $\ell_2$-norm constrained projected gradient ascent attack is indeed equivalent to data-dependent spectral norm regularization.

**Theorem 1.** *For $\epsilon$ small enough such that $\mathcal{B}_\epsilon^2(\mathbf{x}) \subset X(\phi_\mathbf{x})$ and in the limit $\alpha \to \infty$, $\ell_2$-norm constrained projected gradient ascent based adversarial training with a sum-of-squares loss on the logits of the clean and perturbed input $\ell_{\mathrm{adv}}(f(\mathbf{x}), f(\mathbf{x}^*)) = \frac{1}{2}||f(\mathbf{x}) - f(\mathbf{x}^*)||_2^2$ is equivalent to data-dependent spectral norm regularization.*

The proof can be found in Section A.2 in the Appendix.

The conditions on $\epsilon$ and $\alpha$ can be considered specifics of the iteration method. The condition that $\epsilon$ be small enough such that $\mathcal{B}_\epsilon^2(\mathbf{x})$ is contained in the ReLU cell around $\mathbf{x}$ ensures that the Jacobian $\mathbf{J}_{f(\mathbf{x}^*)} = \mathbf{J}_{f(\mathbf{x})}$ for all $\mathbf{x}^* \in \mathcal{B}_\epsilon^2(\mathbf{x})$, while the condition that $\alpha \to \infty$ means that in the update equation for $\mathbf{x}_k$ all the weight is placed on the current gradient direction $\mathbf{v}_k$ whereas no weight is put on the previous iterate $\mathbf{x}_{k-1}$. Note that the limit $\alpha \to \infty$ is well-defined since it is inside the projection operation (the projection of $\mathbf{x}_k$ divides by $\alpha$ again).

Note that, in practice, the Theorem is applicable as long as the Jacobian of the network remains approximately constant in the uncertainty ball under consideration, in which case the correspondence between adversarial training and data-dependent spectral norm regularization holds *approximately* in a region much larger than $X(\phi_\mathbf{x})$. In Section 5, we verify this in the experimental setting.

## 5 EXPERIMENTAL RESULTS

### 5.1 DATASET, ARCHITECTURE & TRAINING METHODS

We trained Convolutional Neural Networks (CNNs) with ReLU activations and batch normalization on the CIFAR10 data set (Krizhevsky & Hinton, 2009). We use a 7-layer CNN as our default platform, since it has good test set accuracy at acceptable computational requirements (we used an estimated 2.5k GPU hours (Titan X) in total for all our experiments). We train each classifier with a number of different training methods: (i) 'Standard': standard empirical risk minimization with a softmax cross-entropy loss, (ii) 'Adversarial': $\ell_2$-norm constrained projected gradient ascent (PGA) based adversarial training with a softmax cross-entropy loss, (iii) 'global SNR': global spectral norm regularization à la Yoshida & Miyato (2017), and (iv) 'd.-d. SNR': data-dependent spectral norm regularization.

Table 1: CIFAR10 test set accuracies and hyper-parameters for the CNN7 and training methods we considered. The regularization constants were chosen such that the models achieve roughly the same test set accuracy on clean examples as the adversarially trained model does.

| Training Method | Accuracy | Hyper-parameters |
|---|---|---|
| Standard Training | 93.5% | — |
| Adversarial Training | 83.6% | $\epsilon = 1.75$, $\alpha = 2\epsilon/\text{iters}$, iters $= 10$ |
| Global Spectral Norm Reg. | 80.4% | $\lambda = 3 \cdot 10^{-4}$, iters=1 |
| Data-dep. Spectral Norm Reg. | 84.6% | $\lambda = 3 \cdot 10^{-2}$, $\epsilon = 1.75$, iters $= 10$ |

As a default attack strategy we use an $\ell_2$-norm constrained PGA white-box attack with cross-entropy adversarial loss $\ell_{\text{adv}}$ and 10 attack iterations. We verified that all our conclusions also hold for larger numbers of attack iterations, however, due to computational constraints we limit the attack iterations to 10. The attack strength $\epsilon$ used for training was chosen to be the smallest value such that almost all adversarially perturbed inputs to the standard model are successfully misclassified, which is $\epsilon = 1.75$ (indicated by a vertical dashed line in the Figures below). The regularization constants of the other training methods were then chosen in such a way that they roughly achieve the same test set accuracy on clean examples as the adversarially trained model does. Further details regarding the experimental setup can be found in Section A.4 in the Appendix. Table 1 summarizes the test set accuracies and hyper-parameters for the training methods we considered. Shaded areas in the plots below denote standard errors with respect to the number of test set samples over which the experiment was repeated.

## 5.2 Spectral Properties

**Effect of training method on singular value spectrum.** We compute the singular value spectrum of the Jacobian $\mathbf{J}_{f(\mathbf{x})}$ for networks $f$ trained with different training methods and evaluated at a number of different test set examples (200 except if stated otherwise). Since we are interested in computing the full singular value spectrum, and not just the dominant singular value and singular vectors as during training, the power method would be too impractical to use, as it gives us access to only one (the dominant) singular value-vector pair at a time. Instead, we first extract the Jacobian (which is per se defined as a computational graph in modern deep learning frameworks) as an input-dim×output-dim dimensional matrix and then use available matrix factorization routines to compute the full SVD of the extracted matrix. For each training method, the procedure is repeated for 200 randomly chosen clean and corresponding adversarially perturbed test set examples. Further details regarding the Jacobian extraction can be found in Section A.5 in the Appendix.

The results are shown in Figure 1 (left). We can see that, compared to the spectrum of the normally trained and global spectral norm regularized model, the spectrum of adversarially trained and data-dependent spectral norm regularized models is significantly damped after training. In fact, the data-dependent spectral norm regularizer seems to dampen the singular values even slightly more effectively than adversarial training, while global spectral norm regularization has almost no effect compared to standard training.

**Alignment of adversarial perturbations with singular vectors.** We compute the cosine-similarity of adversarial perturbations with singular vectors $\mathbf{v}_r$ of the Jacobian $\mathbf{J}_{f(\mathbf{x})}$, extracted at a number of test set examples, as a function of the rank of the singular vectors returned by the SVD decomposition. For comparison we also show the cosine-similarity with the singular vectors of a random network as well as the cosine-similarity with random perturbations.

The results are shown in Figure 1 (right). We can see that for all training methods (except the random network) adversarial perturbations are strongly aligned with the dominant singular vectors while the alignment decreases towards the bottom-ranked singular vectors. For the random network, the alignment is roughly constant with respect to rank. Interestingly, this strong alignment with dominant singular vectors also explains why input gradient regularization and fast gradient method (FGM) based adversarial training do not sufficiently protect against adversarial attacks, namely because the input gradient, resp. a single power method iteration, do not yield a sufficiently good approximation for the dominant singular vector in general.

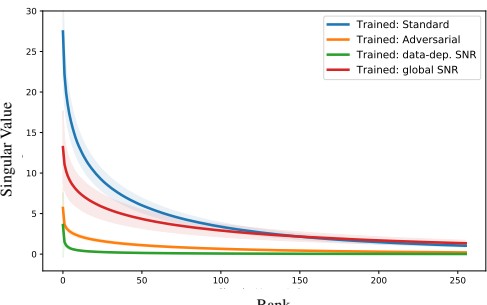 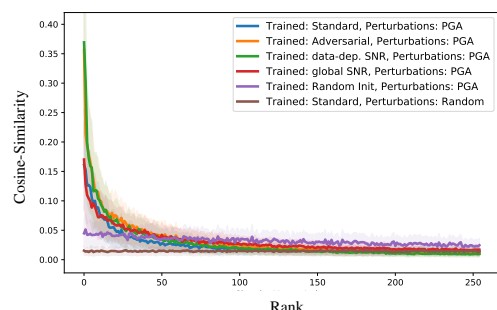

Figure 1: (Left) Singular value spectrum of the Jacobian $\mathbf{J}_{f(\mathbf{x})}$ for networks $f$ trained with different training methods. (Right) Cosine-similarity of adversarial perturbations with singular vectors $\mathbf{v}_r$ of the Jacobian $\mathbf{J}_{f(\mathbf{x})}$, as a function of the rank $r$ of the singular vector. For comparison we also show the cosine-similarity with the singular vectors of a random network as well as the alignment with random perturbations. Curves were aggregated over 200 samples from the test set.

## 5.3 LOCAL LINEARITY

**Validity of linear approximation.** In order to determine the size of the area where a locally linear approximation is valid, we measure the deviation from linearity of $\phi^{L-1}(\mathbf{x} + \mathbf{z})$ as the distance $||\mathbf{z}||_2$ to $\mathbf{x}$ is increased in random and adversarial directions, i.e. we measure $||\phi^{L-1}(\mathbf{x} + \mathbf{z}) - (\phi^{L-1}(\mathbf{x}) + \mathbf{J}_{\phi^{L-1}(\mathbf{x})}\mathbf{z})||_2$ as a function of the distance $||\mathbf{z}||_2$, for random and adversarial perturbations $\mathbf{z}$, aggregated over 200 data points $\mathbf{x}$ in the test set, with adversarial perturbations serving as a proxy for the direction in which the linear approximation holds the least. The purpose of this experiment is to investigate how good the linear approximation for different training methods is, as an increasing number of activation boundaries are crossed with increasing perturbation radius. See Figure 1 in (Raghu et al., 2017) or Figure 3 in (Novak et al., 2018) for an illustration of activation boundary tesselations in the input space.

The results are shown in Figure 2 (left). We can see that adversarial training and data-dependent spectral norm regularization give rise to models that are considerably more linear than the clean trained one, both in random as well as adversarial directions. Compared to the normally trained model, the adversarially trained and spectral norm regularized ones remain flat in random directions for pertubations of considerable magnitude and even remain flat in the adversarial direction for perturbation magnitudes up to the order of the $\epsilon$ used during adversarial training, while the deviation from linearity seems to increase roughly linearly with $||\mathbf{z}||_2$ thereafter. The global spectral norm regularized model behaves similar to the normally trained one.

**Largest singular value over distance.** Figure 2 (right) shows the largest singular value of the linear operator $\mathbf{J}_{f(\mathbf{x}+\mathbf{z})}$ as the distance $||\mathbf{z}||_2$ from $\mathbf{x}$ is increased, both along random and adversarial directions, for different training methods. We can see that the naturally trained network develops large dominant singular values around the data point during training, while the adversarially trained and data-dependent spectral norm regularized models manage to keep the dominant singular value low in the vicinity of $\mathbf{x}$.

## 5.4 ADVERSARIAL ROBUSTNESS

**Adversarial classification accuracy.** A plot of the classification accuracy on adversarially perturbed test examples, as a function of the perturbation strength $\epsilon$, is shown in Figure 3 (left). We can see that the adversarial accuracy of the data-dependent spectral norm regularized model is comparable to that of the adversarially trained model, while global spectral norm regularization does not seem to robustify the model against adversarial attacks. This is in line with our earlier observation that adversarial perturbations tend to align with dominant singular vectors and that adversarial training and data-dependent spectral norm regularization dampen the singular values. Additional results against $\ell_\infty$-PGA attack are provided in Section A.6 in the Appendix. The conclusions for this and the other experiments remain the same. This indicates that the main effect of adversarial training is captured by data-dependent spectral norm regularization.

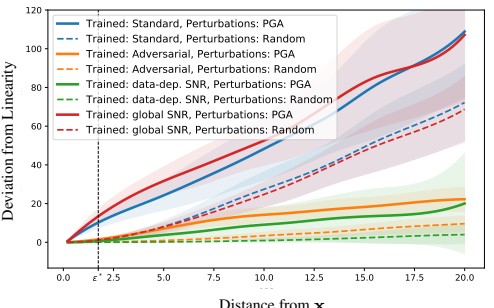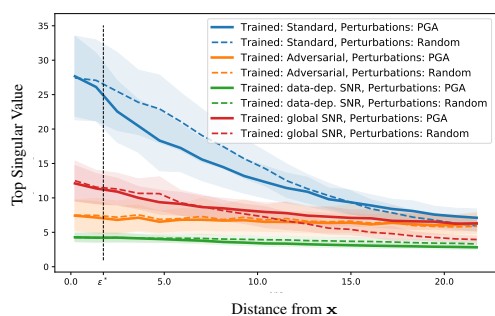

Figure 2: (Left) Deviation from linearity $||\phi^{L-1}(\mathbf{x} + \mathbf{z}) - (\phi^{L-1}(\mathbf{x}) + \mathbf{J}_{\phi^{L-1}(\mathbf{x})}\mathbf{z})||_2$ as a function of the distance $||\mathbf{z}||_2$ from $\mathbf{x}$ for random and adversarial perturbations $\mathbf{z}$. (Right) Largest singular value of the linear operator $\mathbf{J}_{f(\mathbf{x}+\mathbf{z})}$ as a function of the magnitude $||\mathbf{z}||_2$ of random and adversarial perturbations $\mathbf{z}$. The dashed vertical line indicates the $\epsilon$ used during adversarial training. Curves were aggregated over 200 samples from the test set.

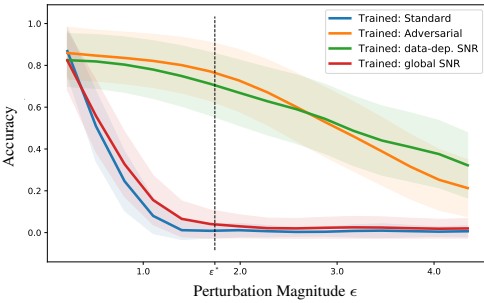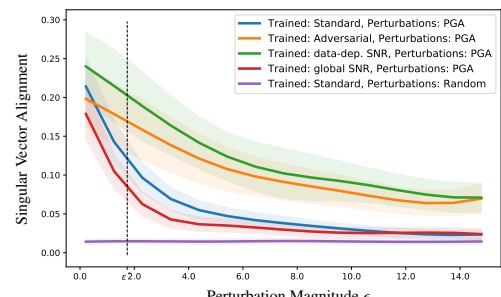

Figure 3: (Left) Classification accuracy as a function of perturbation strength $\epsilon$. (Right) Alignment of adversarial perturbations with dominant singular vector of $\mathbf{J}_{f(\mathbf{x})}$ as a function of perturbation magnitude $\epsilon$. The dashed vertical line indicates the $\epsilon$ used during adversarial training. Curves were aggregated over 2000 samples from the test set.

**Alignment of adversarial perturbations with dominant singular vector.** Figure 3 (right) shows the cosine-similarity of adversarial perturbations of mangitude $\epsilon$ with the dominant singular vector of $\mathbf{J}_{f(\mathbf{x})}$, as a function of perturbation magnitude $\epsilon$. For comparison, we also include the alignment with random perturbations. For all training methods, the larger the perturbation magnitude $\epsilon$, the lesser the adversarial perturbation aligns with the dominant singular vector of $\mathbf{J}_{f(\mathbf{x})}$, which is to be expected for a simultaneously increasing deviation from linearity. The alignment is similar for adversarially trained and data-dependent spectral norm regularized models and for both larger than that of global spectral norm regularized and naturally trained models.

## 6 Conclusion

We established a theoretical link between adversarial training and operator norm regularization for deep neural networks. Specifically, we presented a data-dependent variant of spectral norm regularization that directly regularizes large singular values of a neural network in regions that are supported by the data and proved that it is equivalent to adversarial training based on a specific $\ell_2$-norm constrained projected gradient ascent attack. This fundamental connection confirms the long-standing argument that a network's sensitivity to adversarial examples is tied to its spectral properties and opens the door for adversarially robust generalization bounds via data-dependent spectral norm based ones. We also conducted extensive empirical evaluations showing that (i) adversarial perturbations align with dominant singular vectors, (ii) adversarial training and data-dependent spectral norm regularization dampen the singular values, and (iii) both training methods give rise to models that are significantly more linear around data points than normally trained ones.

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

## A    APPENDIX

### A.1    ROBUST OPTIMIZATION AND REGULARIZATION FOR LINEAR REGRESSION

In this section, we recapitulate the basic ideas on the relation between robust optimization and regularization presented in Bertsimas & Copenhaver (2018). Note that the notation deviates slightly from the main text: most importantly, the perturbations $\triangle$ refer to perturbations of the entire training data $\boldsymbol{X}$, as is common in robust optimization. Consider linear regression with additive perturbations $\triangle$ of the data matrix $\boldsymbol{X}$

$$\min_{\mathbf{w}} \max_{\triangle \in \mathcal{U}} g\left(\mathbf{y} - (\boldsymbol{X} + \triangle)\mathbf{w}\right), \quad \text{with loss} \quad g : \mathbb{R}^n \to \mathbb{R}, \tag{13}$$

where $\mathcal{U}$ denotes the uncertainty set. A general way to construct $\mathcal{U}$ is as a ball of bounded matrix norm perturbations $\mathcal{U} = \{\triangle : \|\triangle\| \leq \lambda\}$. Of particular interest are induced matrix norms

$$\|\mathbf{A}\|_{g,h} := \max_{\mathbf{w}} \left\{ \frac{g(\mathbf{Aw})}{h(\mathbf{w})} \right\}, \quad g : \mathbb{R}^n \to \mathbb{R}, \quad h : \mathbb{R}^d \to \mathbb{R}, \tag{14}$$

where $g$ is a semi-norm and $h$ is a norm. It is obvious that if $g$ fulfills the triangle inequality then one can upper bound

$$g\left(\mathbf{y} - (\boldsymbol{X} + \triangle)\mathbf{w}\right) \leq g(\mathbf{y} - \boldsymbol{X}\mathbf{w}) + g(\triangle\mathbf{w}) \leq g(\mathbf{y} - \boldsymbol{X}\mathbf{w}) + \lambda\,h(\mathbf{w}), \quad \forall \triangle \in \mathcal{U}, \tag{15}$$

by using (a) the triangle inequality and (b) the definition of the matrix norm.

The question then is, under which circumstances both inequalities become equalities at the maximizing $\triangle^*$. It is straightforward to check (Bertsimas & Copenhaver, 2018, Theorem 1) that specifically we may choose the rank 1 matrix

$$\triangle^* = \frac{\lambda}{g(\mathbf{r})}\mathbf{r}\mathbf{v}^\top, \quad \text{where} \ \ \mathbf{r} = \mathbf{y} - \boldsymbol{X}\mathbf{w}\,, \ \ \mathbf{v} = \underset{\mathbf{v}:h^*(\mathbf{v})=1}{\arg\max}\left\{\mathbf{v}^\top\mathbf{w}\right\}, \ \ (h^* \text{ is the dual norm})\,. \tag{16}$$

If $g(\mathbf{r}) = 0$ then one can pick any $\mathbf{u}$ for which $g(\mathbf{u}) = 1$ to form $\triangle = \lambda\mathbf{u}\mathbf{v}^\top$ (such a $\mathbf{u}$ has to exist if $g$ is not identically zero). This shows that, for robust linear regression with induced matrix norm uncertainty sets, robust optimization is equivalent to regularization.

### A.2    PROOF OF MAIN THEOREM

**Theorem 1** *For $\epsilon$ small enough such that $\mathcal{B}_\epsilon^2(\mathbf{x}) \subset X(\phi_{\mathbf{x}})$ and in the limit $\alpha \to \infty$, $\ell_2$-norm constrained projected gradient ascent based adversarial training with a sum-of-squares loss on the logits of the clean and perturbed input $\ell_{\mathrm{adv}}(f(\mathbf{x}), f(\mathbf{x}^*)) = \frac{1}{2}\|f(\mathbf{x}) - f(\mathbf{x}^*)\|_2^2$ is equivalent to data-dependent spectral norm regularization.*

To proof the theorem we need show that the update equations in (12) for $\ell_2$-norm constrained projected gradient ascent based adversarial training with the sum-of-squares loss function and the above conditions on $\epsilon$ and $\alpha$ reduce to the corresponding update equations for data-dependent spectral norm regularization in equation (7).

We start with the following lemma.

**Lemma 1.** *In the limit $\alpha \to \infty$, the projection $\mathbf{x}_k = \Pi_{\mathcal{B}_\epsilon^2(\mathbf{x})}(\mathbf{x}_{k-1} + \alpha\mathbf{v}_k)$ reduces to $\mathbf{x}_k = \mathbf{x} + \epsilon\mathbf{v}_k$.*

*Proof.* The projection $\mathbf{x}_k = \Pi_{\mathcal{B}_\epsilon^2(\mathbf{x})}(\mathbf{x}_{k-1} + \alpha\mathbf{v}_k)$ can be expressed as follows,

$$\mathbf{x}_k = \mathbf{x} + \epsilon(\tilde{\mathbf{x}}_k - \mathbf{x})/\max(\epsilon, \|\tilde{\mathbf{x}}_k - \mathbf{x}\|_2) \quad \text{with} \quad \tilde{\mathbf{x}}_k = \mathbf{x}_{k-1} + \alpha\mathbf{v}_k\,, \tag{17}$$

where the $\max(\epsilon, \|\tilde{\mathbf{x}}_k - \mathbf{x}\|_2)$ ensures that if $\|\tilde{\mathbf{x}}_k - \mathbf{x}\|_2 < \epsilon$ then $\mathbf{x}_k = \tilde{\mathbf{x}}_k$.

Thus,

$$\mathbf{x}_k = \lim_{\alpha \to \infty} \Pi_{\mathcal{B}_\epsilon^2(\mathbf{x})}(\mathbf{x}_{k-1} + \alpha \mathbf{v}_k) \tag{18}$$

$$= \lim_{\alpha \to \infty} \mathbf{x} + \epsilon \frac{\tilde{\mathbf{x}}_k - \mathbf{x}}{\max(\epsilon, ||\tilde{\mathbf{x}}_k - \mathbf{x}||_2)} \tag{19}$$

$$= \mathbf{x} + \epsilon \lim_{\alpha \to \infty} \frac{\mathbf{x}_{k-1} + \alpha \mathbf{v}_k - \mathbf{x}}{\max(\epsilon, ||\mathbf{x}_{k-1} + \alpha \mathbf{v}_k - \mathbf{x}||_2)} \tag{20}$$

$$= \mathbf{x} + \epsilon \lim_{\alpha \to \infty} \frac{\alpha(\mathbf{v}_k + \frac{1}{\alpha}(\mathbf{x}_{k-1} - \mathbf{x}))}{\max(\epsilon, \alpha||\mathbf{v}_k + \frac{1}{\alpha}(\mathbf{x}_{k-1} - \mathbf{x})||_2)} \tag{21}$$

$$= \mathbf{x} + \epsilon \lim_{\alpha \to \infty} \frac{\alpha(\mathbf{v}_k + \frac{1}{\alpha}(\mathbf{x}_{k-1} - \mathbf{x}))}{\alpha||\mathbf{v}_k + \frac{1}{\alpha}(\mathbf{x}_{k-1} - \mathbf{x})||_2} \tag{22}$$

$$= \mathbf{x} + \epsilon \mathbf{v}_k / ||\mathbf{v}_k||_2 \tag{23}$$

$$= \mathbf{x} + \epsilon \mathbf{v}_k \tag{24}$$

where in the third-to-last line we used that the $\max$ will be attained at its second argument in the limit $\alpha \to \infty$ since $||\mathbf{v}_k + \frac{1}{\alpha}(\mathbf{x}_{k-1} - \mathbf{x})||_2 > 0$ for $\mathbf{v}_k \neq 0$, and in the last line we used that $\mathbf{v}_k$ is normalized by construction, i.e. by how it is defined in the update equations in (12).

$\square$

*Proof of Theorem 1.* With the lemma in place, it is now easy to show that for $\epsilon$ small enough, such that $\mathcal{B}_\epsilon^2(\mathbf{x}) \subset X(\phi_\mathbf{x})$, and in the limit $\alpha \to \infty$, adversarial training reduces to data-dependent spectral norm regularization. Let us show this step by step.

In the limit $\alpha \to \infty$, we have for the forward pass

$$\tilde{\mathbf{u}}_k = \nabla_\mathbf{z} \frac{1}{2}||f(\mathbf{x}) - \mathbf{z}||_2^2 \big|_{\mathbf{z}=f(\mathbf{x}_{k-1})} \tag{25}$$

$$= f(\mathbf{x}_{k-1}) - f(\mathbf{x}) \tag{26}$$

$$= \mathbf{J}_{f(\mathbf{x})}(\mathbf{x}_{k-1} - \mathbf{x}) \tag{27}$$

$$= \epsilon \mathbf{J}_{f(\mathbf{x})} \mathbf{v}_{k-1}, \tag{28}$$

where the last equality holds by the lemma.

For the backward pass, we have

$$\tilde{\mathbf{v}}_k = \mathbf{J}_{f(\mathbf{x}_{k-1})}^\top \mathbf{u}_k = \mathbf{J}_{f(\mathbf{x})}^\top \mathbf{u}_k, \tag{29}$$

since $\mathbf{J}_{f(\mathbf{x}_k)} = \mathbf{J}_{f(\mathbf{x})}$ for all $\mathbf{x}_k \in \mathcal{B}_\epsilon^2(\mathbf{x})$.

Together, this gives

$$\begin{aligned} \mathbf{u}_k &\leftarrow \tilde{\mathbf{u}}_k / ||\tilde{\mathbf{u}}_k||_2, \quad \tilde{\mathbf{u}}_k \leftarrow \epsilon \mathbf{J}_{f(\mathbf{x})} \mathbf{v}_{k-1} \\ \mathbf{v}_k &\leftarrow \tilde{\mathbf{v}}_k / ||\tilde{\mathbf{v}}_k||_2, \quad \tilde{\mathbf{v}}_k \leftarrow \mathbf{J}_{f(\mathbf{x})}^\top \mathbf{u}_k \\ \mathbf{x}_k &\leftarrow \mathbf{x} + \epsilon \mathbf{v}_k. \end{aligned} \tag{30}$$

Observing that the first two update equations don't depend on $\mathbf{x}_k$, and absorbing the $\epsilon$ into the normalization of $\tilde{\mathbf{u}}_k$, we precisely end up with the data-dependent spectral norm regularization in Equation (7):

$$\begin{aligned} \mathbf{u}_k &\leftarrow \tilde{\mathbf{u}}_k / ||\tilde{\mathbf{u}}_k||_2, \quad \tilde{\mathbf{u}}_k \leftarrow \mathbf{J}_{f(\mathbf{x})} \mathbf{v}_{k-1} \\ \mathbf{v}_k &\leftarrow \tilde{\mathbf{v}}_k / ||\tilde{\mathbf{v}}_k||_2, \quad \tilde{\mathbf{v}}_k \leftarrow \mathbf{J}_{f(\mathbf{x})}^\top \mathbf{u}_k \end{aligned} \tag{31}$$

We have thus proved that the update equations for the adversarial attack in Theorem 1 correspond exactly to those of data-dependent spectral norm regularization.

It is also easy to see that the adversarial training objective in Equation (10) corresponds to that of data-dependent spectral norm regularization for the specific choice of adversarial loss function

$$\min \theta \to \mathbb{E}_{(\mathbf{x},y) \sim \hat{P}} \left[ \ell(y, f(\mathbf{x})) + \lambda \max_{\mathbf{x}^* \in \mathcal{B}_\epsilon^2(\mathbf{x})} \frac{1}{2}||f(\mathbf{x}) - f(\mathbf{x}^*)||_2^2 \right]. \tag{32}$$

which by the condition $\mathcal{B}_\epsilon^2(\mathbf{x}) \subset X(\phi_\mathbf{x})$ and $\mathbf{x}^* = \mathbf{x} + \epsilon \mathbf{v}$ is equivalent to

$$\min \theta \to \mathbf{E}_{(\mathbf{x},y)\sim\hat{P}}\left[\ell(y, f(\mathbf{x})) + \frac{\lambda\epsilon^2}{2}(\mathbf{J}_{f(\mathbf{x})}\mathbf{v})^2\right] \tag{33}$$

where the data-dependent singular vector $\mathbf{v}$ is computed via Equation (7).

$\square$

## A.3 EFFECT OF THE ADVERSARIAL LOSS FUNCTION ON THE LOGIT-SPACE DIRECTION

The adversarial loss function determines the logit-space direction $\mathbf{u}_k$ of the directional derivative in the power method like formulation of adversarial training in Equation 12.

Let us consider this for the softmax cross-entropy loss, defined as

$$\ell_{\mathrm{adv}}(y, \mathbf{z}) = -\log(s_y(\mathbf{z})) = -\mathbf{z}_y + \log\left(\sum_{k=1}^d \exp(\mathbf{z}_k)\right) \quad , \; s_y(\mathbf{z}) := \frac{\exp(\mathbf{z}_y)}{\sum_{k=1}^d \exp(\mathbf{z}_k)} \tag{34}$$

Untargeted $\ell_2$-PGA on softmax cross-entropy loss: (forward pass)

$$\left[\tilde{\mathbf{u}}_k \leftarrow \nabla_\mathbf{z}\ell_{\mathrm{adv}}(y, \mathbf{z})|_{\mathbf{z}=f(\mathbf{x}_k)}\right]_i = s_i(f(\mathbf{x}_k)) - \delta_{iy} \tag{35}$$

Targeted $\ell_2$-PGA on softmax cross-entropy loss: (forward pass)

$$\left[\tilde{\mathbf{u}}_k \leftarrow -\nabla_\mathbf{z}\ell_{\mathrm{adv}}(y_{\mathrm{adv}}, \mathbf{z})|_{\mathbf{z}=f(\mathbf{x}_k)}\right]_i = \delta_{iy_{\mathrm{adv}}} - s_i(f(\mathbf{x}_k)) \tag{36}$$

Notice that the logit gradient can be computed in a forward pass by analytically expressing it in terms of the arguments of the objective function (this is why we call the $\mathbf{u}_k$ update a forward pass).

Interestingly, for a temperature-dependent softmax cross-entropy loss, the logit-space direction becomes a "label-flip" vector in the low-temperature limit (high inverse temperature $\beta \to \infty$) where the softmax $s_y^\beta(\mathbf{z}) := \exp(\beta\mathbf{z}_y)/(\sum_{k=1}^d \exp(\beta\mathbf{z}_k))$ converges to the argmax: $s^\beta(\mathbf{z}) \xrightarrow{\beta\to\infty} \arg\max(\mathbf{z})$. E.g. for targeted attacks $\left[\mathbf{u}_k^{\beta\to\infty}\right]_i = \delta_{iy_{\mathrm{adv}}} - \delta_{iy(\mathbf{x}_k)}$. This implies that in the high $\beta$ limit, iterative PGA finds an input space perturbation $\mathbf{v}_k$ that corresponds to the steepest ascent of $f$ along the "label flip" direction $\mathbf{u}_k^{\beta\to\infty}$.

**A note on canonical link functions.** Interestingly, the gradient of the loss w.r.t. the logits of the classifier takes the form "prediction - target" for both the sum-of-squares error as well as the softmax cross-entropy loss. This is in fact a general result of modelling the target variable with a conditional distribution from the exponential family along with a canonical link (activation) function. This means that in both cases, adversarial attacks try to find perturbations in input space that induce a logit perturbation that aligns with the difference between the current prediction and the attack target.

## A.4 DATASET, ARCHITECTURE & TRAINING METHODS

We trained Convolutional Neural Networks (CNNs) with seven hidden layers and batch normalization on the CIFAR10 data set Krizhevsky & Hinton (2009). The CIFAR10 dataset consists of 60k $32 \times 32$ colour images in 10 classes, with 6k images per class. It comes in a pre-packaged train-test split, with 50k training images and 10k test images, and can readily be downloaded from https://www.cs.toronto.edu/~kriz/cifar.html.

We conduct our experiments on a pre-trained standard convolutional neural network, employing 7 convolutional layers, augmented with BatchNorm, ReLU nonlinearities and MaxPooling. The network achieves 93.5% accuracy on a clean test set. Relevant links to download the pre-trained model can be found in our codebase.

We adopt the following standard preprocessing and data augmentation scheme: Each training image is zero-padded with four pixels on each side, randomly cropped to produce a new image with the original dimensions and horizontally flipped with probability one half. We also standardize each image to have zero mean and unit variance when passing it to the classifier.

The attack strength $\epsilon$ used for PGA was chosen to be the smallest value such that almost all adversarially perturbed inputs to the standard model are successfully misclassified, which is $\epsilon = 1.75$. The regularization constants of the other training methods were then chosen in such a way that they roughly achieve the same test set accuracy on clean examples as the adversarially trained model does, i.e. we allow a comparable drop in clean accuracy for regularized and adversarially trained models. When training the derived regularized models, we started from a pre-trained checkpoint and ran a hyper-parameter search over number of epochs, learning rate and regularization constants. Table 1 summarizes the test set accuracies and hyper-parameters for all the training methods we considered.

### A.5 EXTRACTING JACOBIAN AS A MATRIX

Since we know that any neural network with its nonlinear activation function set to fixed values represents a linear operator, which, locally, is a good approximation to the neural network itself, we develop a method to fully extract and specify this linear operator in the neighborhood of any input datapoint $\mathbf{x}$. We have found the naive way of determining each entry of the linear operator by consecutively computing changes to individual basis vectors to be numerically unstable and therefore have settled for a more robust alternative:

In a first step, we run a set of randomly perturbed versions of $\mathbf{x}$ through the network (with fixed activation functions) and record their outputs at the particular layer that is of interest to us (usually the logit layer). In a second step, we compute a linear regression on these input-output pairs to obtain a weight matrix $\mathbf{W}$ as well as a bias vector $\mathbf{b}$, thereby fully specifying the linear operator. The singular vectors and values of $\mathbf{W}$ can be obtained by performing an SVD.

### A.6 FURTHER EXPERIMENTAL RESULTS FOR $\ell_\infty$-PGA

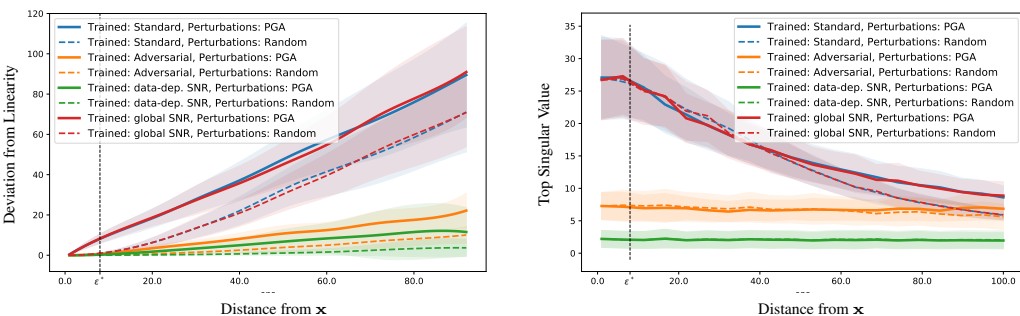

Figure 4: (Left) Deviation from linearity $||\phi^{L-1}(\mathbf{x} + \mathbf{z}) - (\phi^{L-1}(\mathbf{x}) + \mathbf{J}_{\phi^{L-1}(\mathbf{x})}\mathbf{z})||_2$ as a function of the distance $||\mathbf{z}||_2$ from $\mathbf{x}$ for random and $\ell_\infty$-PGA adversarial perturbations $\mathbf{z}$. (Right) Largest singular value of the linear operator $\mathbf{J}_{f(\mathbf{x}+\mathbf{z})}$ as a function of the magnitude $||\mathbf{z}||_2$ of random and $\ell_\infty$-PGA adversarial perturbations $\mathbf{z}$. The dashed vertical line indicates the $\epsilon$ used during adversarial training. Curves were aggregated over 200 samples from the test set.

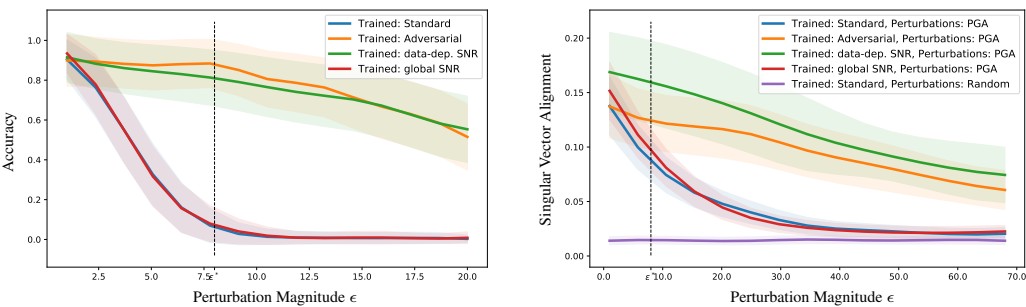

Figure 5: (Left) Classification accuracy as a function of perturbation strength $\epsilon$ (measured in 8-bit). (Right) Alignment of $\ell_\infty$-PGA adversarial perturbations with dominant singular vector of $\mathbf{J}_{f(\mathbf{x})}$ as a function of perturbation magnitude $\epsilon$. The dashed vertical line indicates the $\epsilon$ used during adversarial training. Curves were aggregated over 2000 samples from the test set.

## A.7 FURTHER EXPERIMENTAL RESULTS FOR GLOBAL SNR WITH 10 ITERATIONS

In the main section, we have implemented the baseline version of global SNR as close as possible to the descriptions in Yoshida & Miyato (2017). However, this included a recommendation from the authors to perform only a single update iteration to the spectral decompositions of the weight matrices per training step. As this is computationally less demanding than the 10 iterations per training step spent on both adversarial training, as well as data-dependent spectral norm regularization, we verify that performing 10 iterations makes no difference to the method of Yoshida & Miyato (2017). Figures 6 and 7 reproduce the curves for global SNR from the main part (having used 1 iteration) and overlap it with the same experiments, but done with global SNR using 10 iterations. As can be seen, there is no significant difference between the two.

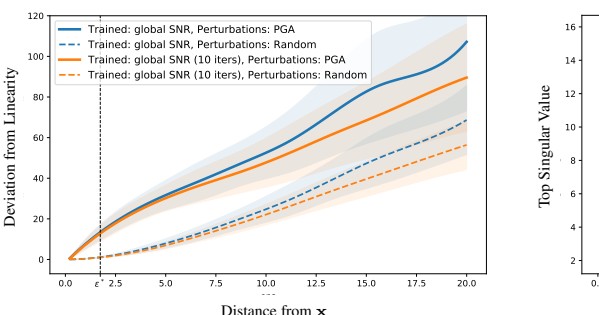 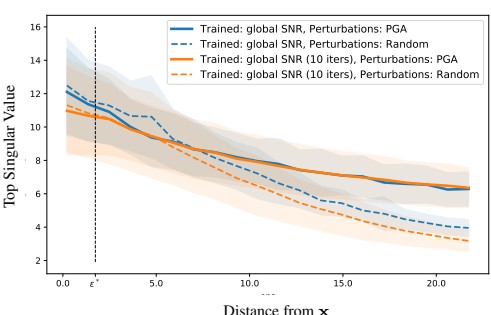

Figure 6: (Left) Deviation from linearity $||\phi^{L-1}(\mathbf{x} + \mathbf{z}) - (\phi^{L-1}(\mathbf{x}) + \mathbf{J}_{\phi^{L-1}(\mathbf{x})}\mathbf{z})||_2$ as a function of the distance $||\mathbf{z}||_2$ from $\mathbf{x}$ for random and adversarial perturbations $\mathbf{z}$. (Right) Largest singular value of the linear operator $\mathbf{J}_{f(\mathbf{x}+\mathbf{z})}$ as a function of the magnitude $||\mathbf{z}||_2$ of random and adversarial perturbations $\mathbf{z}$. The dashed vertical line indicates the $\epsilon$ used during adversarial training. Curves were aggregated over 200 samples from the test set.

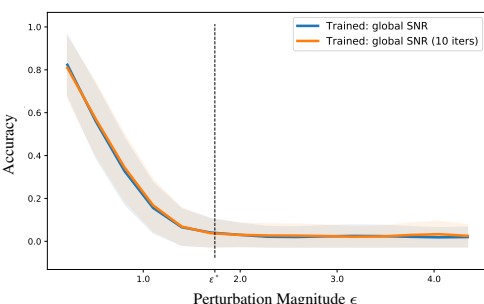 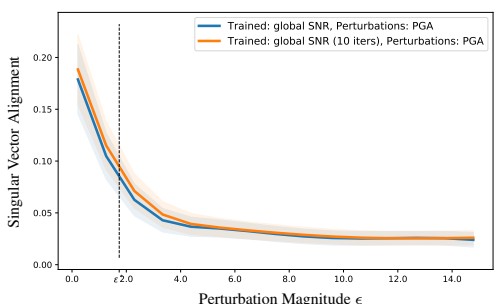

Figure 7: (Left) Classification accuracy as a function of perturbation strength $\epsilon$. (Right) Alignment of adversarial perturbations with dominant singular vector of $\mathbf{J}_{f(\mathbf{x})}$ as a function of perturbation magnitude $\epsilon$. The dashed vertical line indicates the $\epsilon$ used during adversarial training. Curves were aggregated over 2000 samples from the test set.

## A.8 HYPERPARAMETER SWEEP

We tested the following hyperparameter configurations during training and selected the best performing for each training method. Table 2 details the searched parameters.

## A.9 ADVERSARIAL TRAINING WITH LARGE $\alpha$

Figure 8 shows the result of empirically raising the adversarial learning rate during adversarial training with PGA to very large values. As can be seen, the adversarial robustness initially rises, but after some threshold it levels out and does not change significantly even at very large values.

Table 2: Hyperparameter sweep during training.

| TRAINING METHOD | HYPERPARAMETER | VALUES TESTED |
|---|---|---|
| ADVERSARIAL TRAINING | $\epsilon$ | $0.5, 0.75, 1.0, 1.25, 1.5, 1.75, 2.0, 2.25,$ |
| | | $2.5, 2.75, 3.0, 3.25, 3.5, 3.75, 4.0$ |
| | $\alpha$ | $\epsilon/\text{ITERS}, 2\epsilon/\text{ITERS}, 3\epsilon/\text{ITERS}, 4\epsilon/\text{ITERS}, 5\epsilon/\text{ITERS}$ |
| | ITERS | $1, 2, 3, 5, 8, 10, 15, 20, 30, 40, 50$ |
| GLOBAL SPECTRAL NORM REG. | $\lambda$ | $1 \cdot 10^{-5}, 3 \cdot 10^{-5}, 1 \cdot 10^{-4}, 3 \cdot 10^{-4},$ |
| | | $1 \cdot 10^{-3}, 3 \cdot 10^{-3}, 1 \cdot 10^{-2}, 3 \cdot 10^{-2}, 1 \cdot 10^{-1}, 3 \cdot 10^{-1},$ |
| | | $1 \cdot 10^{0}, 3 \cdot 10^{0}, 1 \cdot 10^{1}, 3 \cdot 10^{1}$ |
| | ITERS | $1, 10$ |
| DATA-DEP. SPECTRAL NORM REG. | $\lambda$ | $1 \cdot 10^{-5}, 3 \cdot 10^{-5}, 1 \cdot 10^{-4}, 3 \cdot 10^{-4},$ |
| | | $1 \cdot 10^{-3}, 3 \cdot 10^{-3}, 1 \cdot 10^{-2}, 3 \cdot 10^{-2}, 1 \cdot 10^{-1}, 3 \cdot 10^{-1},$ |
| | | $1 \cdot 10^{0}, 3 \cdot 10^{0}, 1 \cdot 10^{1}, 3 \cdot 10^{1}$ |
| | ITERS | $1, 2, 3, 5, 8, 10, 15, 20, 30, 40, 50$ |

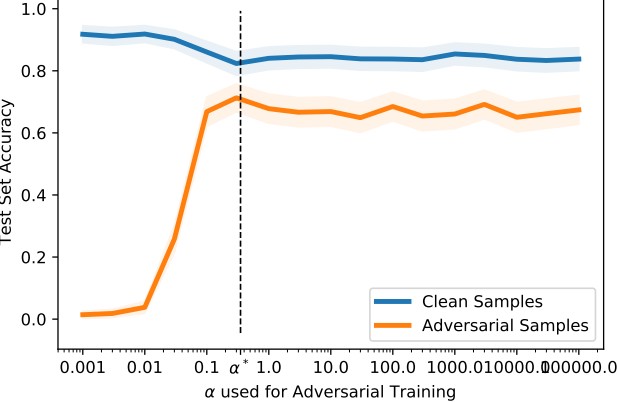

Figure 8: Test set accuracy on clean and adversarial examples after using adversarial (PGA) training with a given step size $\alpha$. The dashed line indicates the $\alpha$ used when generating adversarial examples at test time.

