# OpenReview forum: "Adversarial Training Generalizes Data-dependent Spectral Norm Regularization"
_ICLR.cc/2020/Conference — Reject_

### Official Review · AnonReviewer4 · 2019-10-21
**Official Blind Review #4**

**Rating:** 3

**Review:**

This paper studies the link between adversarial training and the proposed data-dependent operator norm regularization for ReLU network. Under specific conditions, in theory, the authors show the equivalence between the l2 PGD training and the regularization method. Empirical experiments are conducted to support the theory.

While this paper gives interesting observation on both theory and empirical study, I think this paper is not qualified for publishing in ICLR due to the following reasons: (1) limited theoretical results; (2) No significant improvement for practical algorithm;

Main argument:

The main theorem 1 seems to be weak as it is only valid for small perturbation region \epsilon and it is unclear how this assumption is consistent to the practice. It is also unclear how the assumption that \alpha \to \infty influences the practical algorithm.

It would be better to generalize the theorem to other \ell_p attack, instead of just \ell_2.

Discussion of computational complexity of the proposed regularization method compared with PGD is missed.

The adversarial robustness is related to the (local) Lipschitz continuity and many other types of regularization decreases the (local) Lipschitz constant. Could you further give result that distinguishes the proposed norm?

It would be better to give improved algorithm for adversarial training based on the current result. The current contribution for further theoretical is too weak as the main theorem requires strong assumption. And I don’t see significant contribution to empirical algorithm.


Minor
Equ (10) seems not the typical one used and seems not the one studied later.


**Experience Assessment:**

I have read many papers in this area.

**Review Assessment: Checking Correctness Of Derivations And Theory:**

I carefully checked the derivations and theory.

**Review Assessment: Checking Correctness Of Experiments:**

I assessed the sensibility of the experiments.

**Review Assessment: Thoroughness In Paper Reading:**

I read the paper at least twice and used my best judgement in assessing the paper.

---

> ### Author Response · Authors · 2019-11-10
> **Author Response to Review (Part 2 of 2)**
>
> *This is part 2 of 2 of our response*
>
> >> Discussion of computational complexity of the proposed regularization method compared to PGD.
>
> In the last sentence of Section 4.1 we compare the computational complexity of d.d. SNR with that of global (data-independent) SNR.
>
> Compared to PGD, d.d. SNR is equally computationally expensive. As stated in Table 1, both PGD and d.d. SNR were implemented with 10 iterations. Since our main claim is to show equivalence of the two methods, this is a valuable piece of information. We will add a sentence to the paper to emphasize this point.
>
>
> >> Many other types of regularization decreases the (local) Lipschitz constant. Could you further give result that distinguishes the proposed norm?
>
> Indeed there are many works that aim at regularizing the Lipschitz constant. However, these works mostly focus on decreasing the global Lipschitz constant, which corresponds to data-independent SNR and gives only a loose bound on adversarial robustness. We would like to stress that this is different from (and weaker than) our presented data-dependent SNR.
>
> One of the main points in our paper, especially our experiments, is that the Lipschitz regularization of previous work cannot account for / does not explain adversarial robustness (see Section 5.4, Figures 3 (left) and 5 (left)). The data-dependent SNR variant introduced in our paper is a novel and significant generalization and the first type of SNR that is equivalent to AT.
>
>
> >> It would be better to give improved algorithm for adversarial training based on the current result. The current contribution for further theoretical is too weak and I don’t see significant contribution to empirical algorithm.
>
> As we stated above, it is not our goal to improve the practical algorithm of adversarial training, but to show its correspondence to data-dependent SNR. In fact, it would be contrary to our main result to try to improve the practical algorithm.
>
> Other than that, we do not understand what the reviewer means by his or her request. Please elaborate.
>
>
> >> Equ (10) seems not the typical one used and seems not the one studied later.
>
> We do study this equation for p=2. See also Equations (33) and (34) in the Appendix, where we show that Equation (10) reduces to Equation (7) under the conditions of our Theorem.
>
> Perhaps the reviewer refers to the setting in which the network is trained to only minimize the adversarially perturbed empirical loss. It is however customary to train the network to minimize a convex combination of a clean empirical loss and an adversarially perturbed empirical loss, see the equation on page 5 in Goodfellow et al. “Explaining and harnessing adversarial examples”.
>
>
> In conclusion, we agree that our Theorem makes strong assumptions, but we believe that 1. it is valuable to be on record and theoretically confirm this long-standing hypothesis and 2. we show extensively that the claim of the Theorem holds well beyond its assumptions in practice. As for improving over AT in the practical sense, we never claim to do so, and it would actually run contrary to our claim.
>
> We hope that these comments provide clarification and we look forward to continuing the discussion.

---

> ### Author Response · Authors · 2019-11-10
> **Author Response to Review (Part 1 of 2)**
>
> *This is part 1 of 2 of our response*
>
> We would like to thank the reviewer for these comments. We hope that with our detailed answers below we can initiate a fruitful discussion that resolves all concerns.
>
> General Concerns:
>
> (i) limited theoretical results
>
> Note that while our Theorem establishes an exact equivalence for sufficiently small epsilon, as noted by the reviewer, we verified in extensive experiments that in practice the correspondence between AT and d.d. SNR holds up very well in a region much larger than proven in our Theorem, specifically large enough to cover common use cases. Therefore, the conclusions of our theory are directly relevant to practitioners.
>
> (ii) No significant improvement for practical algorithm
>
> We would like to stress that we never claimed that d.d. SNR outperforms AT. The main point of our Theorem (indeed our paper) is that there is a correspondence between the two. In fact, it would be contrary to our main claim if one would outperform the other, be that in terms of final adversarial robustness or computational complexity. We believe that our paper shows this equivalence in theory and practice and thereby increases the understanding of adversarial robustness.
>
>
> Specific Concerns:
>
> >> Main theorem is only valid for small perturbations, unclear how this assumption relates to practice.
>
> The condition on epsilon in the Theorem guarantees that the Jacobian is fixed for all x* with ||x* - x|| <= epsilon, in which case the correspondence between \ell_2 norm constrained PGA-based AT and data-dependent SNR was proven to be an exact equivalence.
>
> In practice, however, the correspondence between AT and d.d. SNR holds approximately in a region much larger than proved in the Theorem: As shown in Figure 2 (left) and discussed in Section 5.3 “Validity of linear approximation”, we verified that the Jacobian is almost constant in a region that is roughly the size of the epsilon*-ball used during adversarial training (epsilon* = 1.75 >> epsilon in Theorem).
>
> The correspondence is in fact consistently supported by all our experiments. In Section 5.4 Adversarial Robustness, for instance, we show that a network trained with d.d. SNR is equally robust to adversarial perturbations with varying magnitude as the PGA trained network is.
>
> In other words, the Theorem is applicable in practice as long as the Jacobian of the network remains approximately constant in the epsilon-ball under consideration. We will add a paragraph below the Theorem to make this clear.
>
>
> >> It is unclear how the assumption \alpha \to \infty influences the practical algorithm.
>
> We elaborate on the condition on \alpha in the paragraph below Theorem 1: “in the update equation for x_k all the weight [if \alpha -> \infty] is placed on the current gradient direction v_k whereas no weight is put on the previous iterate x_{k−1}”. Note that because of the projection operator, the limit case is well defined.
>
> Mathematically, lim_{\alpha \to \infty} \Proj ( x_{k-1} + \alpha*v_k ) is equivalent to lim_{\alpha \to \infty} \Proj ( 1/alpha*x_{k-1} + v_k ).
> Therefore, in the practical algorithm, instead of letting the prefactor \alpha in front of v_k go to \infty, we can equivalently let the prefactor 1/alpha in front of x_{k-1} go to zero, see Equations (18)-(24) in Appendix 7.2 “Proof of Main Theorem”.
>
> The key insight of our experiments Section is that there is no significant difference between adversarial training with small \alpha and data-dependent spectral norm regularization (corresponding to AT with \alpha -> \inty). Both have a similar regularizing effect on the spectrum, similar local linearity, similar adversarial robustness. This supports our claim that the effect of AT is captured by d.d. SNR.
>
>
> >> Generalize theorem to other \ell_p norms.
>
> This is a non-trivial generalization that we are currently investigating for a future publication.

---

> ### Author Response · Authors · 2019-11-15
> **Amended plot for large alpha**
>
> Dear Reviewer
>
> In response to your comment wondering what happens in the practical (PGA) algorithm when \alpha goes to infinity, we empirically tested that the effect of Adversarial Training remains constant when provided with consecutively larger \alpha-values. Please refer to the last plot in the Appendix in the revised version of our paper.
>
> Please also have a look at our response to your review below.
> Thank you for your time.

---

### Official Review · AnonReviewer3 · 2019-10-29
**Official Blind Review #2**

**Rating:** 8

**Review:**

This largely theretical paper establishes a theoretical link between adversarial training and operator norm regularization
for DNNs. It is well written and structured, and it falls squarely within the the remit of the conference. The experimental apparatus is thorough and the derivations, proofs and the maths at large seem sound to me, even if I have not checked them in full detail. The study delivers a data-dependent variant of spectral norm regularization affecting large singular values of the DNN. It is proved to be equivalent to adversarial training based on a type of norm-constrained projected gradient ascent attack.
Results are novel and relevant and, in my opinion, they merit acceptance.

**Experience Assessment:**

I have read many papers in this area.

**Review Assessment: Checking Correctness Of Derivations And Theory:**

I assessed the sensibility of the derivations and theory.

**Review Assessment: Checking Correctness Of Experiments:**

I assessed the sensibility of the experiments.

**Review Assessment: Thoroughness In Paper Reading:**

I read the paper at least twice and used my best judgement in assessing the paper.

---

> ### Author Response · Authors · 2019-11-10
> **Author Response to Review**
>
> Dear Reviewer
> We would like to thank you for your comments.
> Your feedback is highly appreciated.

---

### Official Review · AnonReviewer2 · 2019-11-06
**Official Blind Review #2**

**Rating:** 1

**Review:**

Adversarial training generalizes data-dependent spectral norm regularization

This paper shows that, projected gradient descent based adversarial training is similar to the data-dependent spectral norm regularization, and under very restrictive condition, the authors show that this two methods are the same. Some experiments are conducted to support the theory.

Overall, I think this paper is marginal, while the experiments are not convincing. First, the relation between spectral normalization and adversarial training have been investigated by [1], while the fast computational of maximum singular value with power methods have also been proposed in [1]. The authors only give a data-dependent version of the spectral normalization based on the Jacobian of the neural networks, which I think is somewhat weak. The experiments are limited with specific settings that are not generally used in practice, which alleviate my confidence on this paper’s results. Also, the experiment section contain several not so important information. I think the authors should do far more experiments to support the main claim, while move these additional justification to the appendix.

Detailed comments:
1. I think the claim of theorem 1 is somewhat ambiguous. How to guarantee there exists such epsilon satisfies this condition? Is this the case we face in the real world? What will happen if \alpha is not sufficiently large? If we don’t use logits pairing and \ell_2 norm constraint, will the claim hold? I think the correlation behinds the spectral norm and adversarial training is well investigated and use this correlation as the intuition behinds work is enough. This theorem cannot convince me that the proposed methods have a strong theoretical basis.
2. Generally, the neural networks have a large number of parameters (~ millions) for image classification task. The global spectral norm regularization only needs to calculate the spectral norm of each layer’s weight matrix, whose computational cost is acceptable. However, to calculate the Jacobian and use the power methods, we will additionally do several forward pass and backward pass just as adversarial training. As a regularization technique, is this calculation tolerable? If this is some variant of the adversarial training, I don’t find the experiment results support the claim that it will outperform the adversarial training consistantly.
3. Why don’t use some standard neural network architecture like ResNet? As this results is not comparable to other existing work, I’m not sure if this result is meaningful. Also, are the comparisons fair? For example, the regularization coefficient of global spectral norm regularization and data-dependent spectral norm regularization are far more different. And the authors use only 1 iteration to calculate the singular value in global spectral norm regularization, why to do that? Also, what’s the result compared with \ell_p norm constraint adversarial training?
4. The evaluation of some assumption on the network is better moved to appendix, as this is only some sanity check, not the core contribution. More experiments with ResNet, WideResNet, MobileNet etc. on CIFAR100 and ImageNet are more convincing.
5. What’s the attack method in the main context?
6. I think the discussion in Appendix A.5 is somewhat confusing. If the authors want to argue that the network is locally linear so that we can approximate with linear regression, why should we use the power methods?

Still, I feel the contribution of this paper is somewhat weak. I don’t see any improvements of the proposed algorithms compared with the standard adversarial training, as well as the theoretical contribution like adversarial generalization. The experiments are not convincing, as the setting is different from the general setting the community used in adversarial training. I’m not familiar with the results in global spectral normalization and it’s possible that the global spectral normalization may have little gain in adversarial robustness, but in my opinion, the main contribution [1] is the generalization analysis of spectral normalized adversarial trained neural networks, which this paper lacks. On the empirical side, the computation efficiency and performance of the proposed algorithms don’t outperform adversarial training much. So I tend to reject this paper.


[1] Farnia, Farzan, Jesse Zhang, and David Tse. "Generalizable Adversarial Training via Spectral Normalization." International Conference on Learning Representations, 2019.

**Experience Assessment:**

I have read many papers in this area.

**Review Assessment: Checking Correctness Of Derivations And Theory:**

I carefully checked the derivations and theory.

**Review Assessment: Checking Correctness Of Experiments:**

I assessed the sensibility of the experiments.

**Review Assessment: Thoroughness In Paper Reading:**

I read the paper thoroughly.

---

> ### Author Response · Authors · 2019-11-10
> **Author Response to Review (Part 3 of 3)**
>
> *This is part 3 of 3 of our response*
>
> >> 3. Why don’t use some standard neural network architecture like ResNet? Also, are the comparisons fair? For example, the regularization coefficient of global SNR and d.d. SNR are different. And the authors use only 1 iteration to calculate the singular value in global spectral norm regularization, why to do that?
>
> “The regularization constants were chosen such that the models achieve roughly the same test set accuracy on clean examples as the adversarially trained model does.” as was clearly stated in our paper. Hence, yes, the comparisons are fair.
>
> For global SNR, we try to stay as close as possible to the original authors' suggestions. Yoshida & Miato write “One [power method] iteration [per parameter update] was adequate in our experiments” and “we performed only one [power method] iteration [per parameter update] because it was adequate for obtaining a sufficiently good approximation”. Note, the computation of the data-independent regularizer decouples from the empirical loss, hence the power-method iterations can be amortized across data-points.
>
> As stated above, our network architecture is standard, is publically available and is used throughout research.
>
>
> >> 4. The evaluation of some assumption on the network is better moved to appendix, as this is only some sanity check, not the core contribution. More experiments with ResNet, WideResNet, MobileNet etc. on CIFAR100 and ImageNet are more convincing.
>
> Firstly, it is unclear what evaluation of assumptions the reviewer is referring to. Secondly, we sincerely do not expect to see any difference regarding the correspondence “AT <-> d.d. SNR” on other architectures / data sets. Our theorem proves that \ell_2-norm constrained PGA-based AT and d.d. SNR are equivalent for small enough epsilon, while our extensive experiments show that in practice, the correspondence between AT and d.d. SNR holds approximately in a region much larger than proved in the Theorem, the region being roughly the size of the epsilon*-ball used during adversarial training (epsilon* = 1.75 >> epsilon in Theorem), see Figure 2 (left) and discussion in Section 5.3 “Validity of linear approximation”.
>
> Sure, more experiments can always be requested, but we believe that confirming our main claims in practice is more important and valuable than including one further architecture or dataset. Please also consider our comments from the "general comments" section at the beginning of this review on this topic.
>
>
> >> 5. What’s the attack method in the main context?
>
> We evaluated against \ell_2-norm constrained PGA in the main text, as stated in Section 5.1 and Table 1. Additional results for \ell_\infty PGA attack are provided in the Appendix.
>
>
> >> 6. The discussion in Appendix A.5 is somewhat confusing. If the authors want to argue that the network is locally linear so that we can approximate with linear regression, why should we use the power methods?
>
> We use the power method during training, since we only need access to the dominant singular vector. In the experiment section, we more generally study the spectral properties of the Jacobian, requiring us to compute the full spectrum and not just the dominant singular value / vector pair. The full spectral decomposition requires much more computation and is only viable when evaluating / investigating certain properties, not during training. We very clearly stated this in the first paragraph of Section 5.1.
>
>
> >> "Still, I feel the contribution of this paper is somewhat weak. I don’t see any improvements of the proposed algorithms compared with the standard adversarial training, as well as the theoretical contribution like adversarial generalization. The experiments are not convincing, as the setting is different from the general setting the community used in adversarial training. I’m not familiar with the results in global spectral normalization and it’s possible that the global spectral normalization may have little gain in adversarial robustness, but in my opinion, the main contribution [1] is the generalization analysis of spectral normalized adversarial trained neural networks, which this paper lacks. On the empirical side, the computation efficiency and performance of the proposed algorithms don’t outperform adversarial training much. So I tend to reject this paper."
>
> Again, we 1. do not claim to outperform AT, we claim to show its correspondence to d.d. SNR and 2. we show this correspondence in a theoretical way that no previous work has managed to establish. Our experimental section reflects and supports these points very well. Also, we are not in a competition with [1].

---

> ### Author Response · Authors · 2019-11-10
> **Author Response to Review (Part 2 of 3)**
>
> *This is part 2 of 3 of our response*
>
> >> Detailed comments:
>
> >> 1. I think the claim of theorem 1 is somewhat ambiguous. How to guarantee there exists such epsilon satisfies this condition? What will happen if \alpha is not sufficiently large? If we don’t use logits pairing and \ell_2 norm constraint, will the claim hold? I think the correlation behinds the spectral norm and adversarial training is well investigated and use this correlation as the intuition behinds work is enough.
>
> First, we disagree strongly that this is well investigated and that using the correlation as an intuition is enough. Just because previous work has measured a correlation between two things does not mean further research is unnecessary. Establishing the reason behind such a correlation in a formal manner is a definite step forward in our understanding of this phenomenon. None of the previous work has been able to make this theoretical link - or even the necessary insistence that data-dependence is needed to establish this connection.
>
> To address the specific questions.
> Since every datapoint exhibits some configuration of activations of the nonlinearities in the network, the existence of the required epsilon-ball is guaranteed by definition.
> The requirement of alpha to be large is a weak assumption and one that is well-defined because of the projection operator. We took account of this in the paragraph following the theorem. That being said, for finite alpha, the theorem will not hold exactly, but approximately. This is why our experimental section is geared towards showing empirically (with alpha < infinity) what the theorem claims theoretically (for alpha -> infinity).
> Extending this work to other settings, such as other \ell_p norms is a non-trivial extension to this work, which we are currently investigating.
>
> >> 2. The global SNR only needs to calculate the spectral norm of each layer’s weight matrix, whose computational cost is acceptable. However, to calculate the Jacobian and use the power methods, we will additionally do several forward pass and backward pass just as AT. As a regularization technique, is this calculation tolerable? If this is some variant of AT, I don’t find the experiment results support the claim that it will outperform AT consistantly.
>
> We stress again that we never aimed at, nor claimed that, our method outperforms AT. In fact, our experiments show that they are on par, supporting our Theorem that there is a correspondence between the two.
>
> The reviewer is correct that data-dependent SNR is computationally more costly than global SNR, which we state in our paper. In detail, one power-method iteration of d.d. SNR is as costly as one power-method iteration of global SNR, as they involve the same number of matrix-vector products. The reason why d.d. SNR is ultimately a constant factor more costly compared to global SNR is because in global SNR the computation of the regularizer is data-independent (it decouples from the empirical loss), hence the power-method iterations can be amortized across data-points.
> That said, data-dependent SNR is equally costly as PGD-based AT. This again supports our claim that the two correspond to each other and as such, yes, the calculation is tolerable.

---

> ### Author Response · Authors · 2019-11-10
> **Author Response to Review (Part 1 of 3)**
>
> *This is part 1 of 3 of our response*
>
> Before we begin to address the individual comments, we would like to emphasize that the majority of them are already addressed in our paper, including a citation to Farzan et al. [1], which the reviewer refers to multiple times. Most importantly, Farzan et al. [1] (among many others) study “spectral normalization of the DNN’s weight matrices”, i.e. data-INdependent SNR (similar to Yoshida & Miyato - which they also cite and which we have implemented in our paper).
>
> We would like to stress that this is fundamentally different from what we do, which is data-dependent SNR. Notably, data-INdependent SNR can only establish and minimize a loose upper bound on the data-dependent spectral norm. In particular, it cannot account for / does not explain adversarial robustness (see Section 5.4, Figures 3 (left) and 5 (left)).
>
> Our data-dependent variant, on the other hand, is a much stronger regularizer. In fact, we show equivalence with adversarial training, which none of the other previous works can establish. Hence, the data-dependent SNR introduced in our paper is not simply a competing method, but a significant generalization.
>
> From reading the reviewer's comments, we believe that we may not have put enough emphasis in our paper to make this distinction clear. We hope that through this discussion, we can resolve these concerns. We will improve the writing in our paper accordingly.
>
> >> The authors only give a data-dependent version of SNR based on the Jacobian of the neural network, which I think is somewhat weak.
>
> Note that our presented d.d. SNR is a stronger notion of SNR than any of the previous works, including [1], which the reviewer refers to.
> We show this in theory, as our method can be proven to be equivalent to AT under certain conditions, while previous work cannot do that. And we explicitly show in practice that spectral normalization of the DNN’s weight matrices, as studied by [1], cannot account for adversarial robustness (see Figures 3 (left) and 5 (left)), whereas our data-dependent SNR variant does.
>
> >> The fast computational of maximum singular value with power methods have been proposed in [1].
>
> The power-method based computation of dominant singular values has been known for almost a century now (von Mises 1929). And even in the context of adversarial robustness, it has been studied before [1], see e.g. Yoshida & Miato (which is also cited by [1]).
>
> As such, we do not claim power method based regularization of singular values to be a novel contribution. However, we are the first to provide a power method based formulation of AT and establish a theoretical equivalence between AT and d.d. SNR.
>
> >> The experiments are limited with specific settings that are not generally used in practice. Also, the experiment section contain several not so important information.
>
> Our model architecture and hyperparameter settings come from publically available and widely used settings and reach comparable performance to state-of-the-art models, while still being feasible to do research on without huge resource requirements.
>
> Note that our goal is not to outperform adversarial training, but to show its equivalence to d.d. SNR. We believe our experimental section does show this very thoroughly. Further, what the reviewer calls "not so important information" is actually extremely vital, since it shows that the conditions of our Theorem are well fulfilled in practice. If we wanted to suggest a new method for robustifying networks and improve over adversarial training, the reviewer would be entirely correct and the experimental section should look very different, but that would be contrary to our main Theorem. Our experimental section is aimed at showing that d.d. SNR corresponds to AT in a practical setting and that it does so in accordance with our theory, as we empirically confirm the conditions necessary for our theory to hold.
>
> We have received and continue to receive praise for the thorough experimental evaluation in this paper from other researchers, precisely because it achieves what it is supposed to achieve. Hopefully, given this new perspective, the choice of our experiments makes more sense.

---

> > ### Comment · AnonReviewer2 · 2019-11-10
> > **Thanks for your response. Below are my ideas.**
> >
> > Regarding the theoretical analysis:
> > By saying that the claim of theorem 1 is ambiguous, I mean, it’s not precise. Can you give some precise condition for epsilon and alpha, like the upper bound of epsilon and lower bound of alpha for your theorem to hold even under some idealized case like two-layer ReLU network? The current way of presentation is NOT a theorem, but a proposition or intuition.
> >
> > Regarding the core contribution and the experiments:
> > In my opinion, I haven’t found the information that this paper can take to the whole community. Like, for example, for practitioner, there is no motivations to use the current methods instead of the adversarial training, as there are several existing efficient implementations of adversarial training that the authors claimed not worse than the proposed methods. For theorists, this paper brought no idea on how to inspire and improve the theory on adversarial training and robust generalization. I expected the authors to achieve either of it, but the theories in this paper DO NOT mean to solve the theoretical problem I mentioned, and I think the proposed methods DO NOT outperform the existing baseline methods on both the efficiency and the accuracy. So I tend to reject this paper.
> >
> > For the claim on the experiments, as the concerns I mentioned, if this methods do not consistently show the improvement of the current methods on the existing state-of-the-art methods, I don’t find the meaning of using such of the methods. For state-of-the-art methods, I would like to see some of the top methods in https://paperswithcode.com/sota/image-classification-on-cifar-10. 93.5% clean accuracy does not mean state-of-the-art performance to me. Also, most of the experiments aims to show that the condition to show the equivalence between the proposed methods and adversarial training can be satisfied in some real world applications and give some interesting empirical observations.
> >
> > For the detailed feedback on the experiments:
> > I miss some of the claims in Section 5.2, so I misunderstood the purpose of Appendix A.5. Sorry for that.
> >
> > For the fairness claim, the authors claimed to tune the hyper-parameters to make sure the regularization methods have the similar test accuracy. Is this shown means the result in Table 1? I tend to have as little hyper-parameters as possible. The current hyper-paremeter setting seems weird to me. Also, if 1 iteration and 10 iterations performs the same, I prefer to use 10 iterations to eliminate the potential question.
> >
> > In summary, due to the reason I mentioned, I tend to reject this paper. But if all of the other reviewers think this should be accepted, I will follow their ideas.

---

> > > ### Author Response · Authors · 2019-11-13
> > > **Author Response to Comment (Part 2 of 2)**
> > >
> > > *This is part 2 of 2 of our response*
> > >
> > > >> For the claim on the experiments, as the concerns I mentioned, if this methods do not consistently show the improvement of the current methods on the existing state-of-the-art methods, I don’t find the meaning of using such of the methods.
> > > For state-of-the-art methods, I would like to see some of the top methods in https://paperswithcode.com/sota/image-classification-on-cifar-10.
> > > 93.5% clean accuracy does not mean state-of-the-art performance to me.
> > > Also, most of the experiments aims to show that the condition to show the equivalence between the proposed methods and adversarial training can be satisfied in some real world applications and give some interesting empirical observations.
> > >
> > > We would like to stress again that our goal is not to outperform any existing methods. In fact, it would be contrary to our main claim if ddSNR would outperform AT. Instead, our goal is to verify the relevance of our Theorem in a practical setting, by empirically confirming the conditions necessary for the correspondence between AT and ddSNR to hold. By establishing this correspondence in theory and practice, our paper contributes significantly to the understanding of adversarial robustness.
> > >
> > > More experiments can of course always be requested, but we believe that confirming our main claims in practice is more important and valuable than including further architectures.
> > > Is there a reasonable expectation why there should be a qualitative difference in experimental results between our 93.5%-accurate model and one that is, say, 96% accurate? Our goal is not to improve the state-of-the-art in adversarial robustness, but to compare all methods fairly on a platform that performs competitively, even if it's not at the very top of the leaderboard.
> > > The CNN architecture we used is similar to the one used in the prominent works of Carlini & Wagner’s “Towards evaluating the robustness of neural networks”, In Security and Privacy (SP), IEEE, 2017 and Papernot et al.’s “Distillation as a defense to adversarial perturbations against deep neural networks”, In Security and Privacy (SP), IEEE, 2016, both reporting a clean test accuracy of 80.9% for standard training without data-augmentation.
> > >
> > > Our accuracies (on clean test) after AT / SNR (~83%) match the ones in related papers that use the architectures requested by the reviewer, e.g. 79% (AlexNet), 83% (ResNet) in Farzan et al. [1] (which the reviewer referred to multiple times), or 79% (ResNet), 87% (WideResNet) in Madry et al. “Towards Deep Learning Models Resistant to Adversarial Attacks”.
> > >
> > >
> > > >> For the detailed feedback on the experiments:
> > > I miss some of the claims in Section 5.2, so I misunderstood the purpose of Appendix A.5. Sorry for that.
> > >
> > > Thank you for the feedback. We are glad to hear that we could clarify these misunderstandings.
> > >
> > >
> > > >> For the fairness claim, the authors claimed to tune the hyper-parameters to make sure the regularization methods have the similar test accuracy. Is this shown means the result in Table 1? I tend to have as little hyper-parameters as possible. The current hyper-paremeter setting seems weird to me. Also, if 1 iteration and 10 iterations performs the same, I prefer to use 10 iterations to eliminate the potential question.
> > >
> > > The purpose of Table 1 is to facilitate reproducibility of our results. Table 1 summarizes the hyperparameters we have found following our protocol to choose the regularization constants such that the models achieve roughly the same test set accuracy on clean examples as the adversarially trained model does.
> > >
> > > For each training method we did a sweep over a relatively broad range of hyper-parameters and the numbers we report represent the best configurations for each of the training methods. For your convenience, we have amended the appendix to include a table of searched values for each method.
> > >
> > > For global SNR à la Yoshida & Miayto, our aim was to stay as close as possible to the authors' suggestions, as indicated already in our previous reply. For your convenience, we have repeated the main experiments with global SNR using 10 iterations instead of one. Please find the plots in the updated appendix. There is no difference between the 1- and 10-iteration versions.

---

> > > ### Author Response · Authors · 2019-11-13
> > > **Author Response to Comment (Part 1 of 2)**
> > >
> > > *This is part 1 of 2 of our response*
> > >
> > > Thank you very much for engaging with our comments, we highly appreciate this.
> > > Please find our comments below.
> > >
> > >
> > > >> Regarding the theoretical analysis:
> > > The claim of theorem 1 is ambiguous, I mean, it’s not precise. Can you give some precise condition for epsilon and alpha, like the upper bound of epsilon and lower bound of alpha for your theorem to hold even under some idealized case like two-layer ReLU network? The current way of presentation is NOT a theorem, but a proposition or intuition.
> > >
> > > We respectfully disagree. The conditions in Theorem 1 *are* precise. Namely, Theorem 1 states that there is an exact correspondence between AT and ddSNR if (i) Bε(x) ⊂ X(φ_x), i.e. if the epsilon-ball is contained in the ReLU cell X(φ_x) around x and (ii) if \alpha \to \infty, i.e. if in the update equation for x_k all the weight is placed on the current gradient direction v_k whereas no weight is put on the previous iterate x_{k−1}, as was clearly stated in the paper.
> > > Also our Theorem does not hold only in some idealized case, it holds in *any* ReLU network.
> > > And for this statement, we provide a formal proof. How is this not a theorem?
> > >
> > > What we believe the reviewer might have in mind instead is an “extended proof” for the “approximate correspondence” between AT and ddSNR.
> > >
> > > Such an “extended proof” would ultimately boil down to introducing a tolerance parameter, say \delta, deriving a radius r such that for all x* with || x* - x || < r, the Jacobian J_f(x*) is “\delta-close” to J_f(x), and then proving that the correspondence between AT and ddSNR holds \delta’-approximately (where \delta’ depends on \delta).
> > >
> > > Proving such an extension is highly non-trivial (one would have to take into account how much “nearby” Jacobians can change based on the crossing of ReLU boundaries) and thus out-of-scope of the current paper. We instead opted to verify this “approximate correspondence” experimentally, showing that in practice, the correspondence between AT and d.d. SNR holds approximately in a region much larger than proved in the Theorem, as already discussed in our previous reply.
> > >
> > >
> > > >> Regarding the core contribution and the experiments:
> > > In my opinion, I haven’t found the information that this paper can take to the whole community. Like, for example, for practitioner, there is no motivations to use the current methods instead of the adversarial training. For theorists, this paper brought no idea on how to inspire and improve the theory on adversarial training and robust generalization. I expected the authors to achieve either of it, but the theories in this paper DO NOT mean to solve the theoretical problem I mentioned, and I think the proposed methods DO NOT outperform the existing baseline methods on both the efficiency and the accuracy.
> > >
> > > Here is what a practitioner could ask themselves: Should I go through the effort to add data-dependent spectral norm regularization in addition to adversarial training on my network to make it more robust? Thanks to us, they now know that this won’t be very fruitful because the two methods do the same thing.
> > >
> > > For theoreticians, the case is even more clear: There have been numerous papers that have shown an empirical correlation between spectral norm regularization and adversarial robustness, yet none of them managed to make a clear formal connection between the two. We establish the first direct proof of this connection.
> > >
> > > Lastly, again, we never claim that our methods outperform AT, or that they are in any way preferrable. We also do not claim to solve the learning theory problem of deriving adversarially robust generalization bounds, although we do believe that the correspondence we establish opens the door for such bounds via generalizations of existing global spectral norm based ones [e.g. Bartlett et al. “Spectrally-normalized margin bounds for neural networks” or Neyshabur et al. “Norm-based capacity control in neural networks”] to our new notion of data-dependent spectral norm regularization.

---

> > > ### Author Response · Authors · 2019-11-15
> > > **Additional Results**
> > >
> > > Dear Reviewer
> > > In response to your comments - and in parallel to formulating our response - we had started replicating our experiments on a WResNet architecture. In clean training, this reaches 96.3% accuracy on CIFAR10 after 200 epochs of training. We hoped that this platform would satisfy the reviewer's request for evaluation on state-of-the-art models.
> > > However, due to the high computational requirements of this model, our evaluations did not finish in time for this rebuttal period. Specifically, for both AT and ddSNR we only managed to perform 95 epochs, leaving the model training in an unfinished state.
> > > We do not want to include half-done results in our paper, but we feel it contributes to the discussion if we state here that as of now, there appears to be no significant difference in adversarial robustness between the adversarially trained variant and a model trained using ddSNR (that has an equivalent drop in clean accuracy as the one trained using AT). This is further evidence in favor of our main claim.
> > >
> > > Accuracy on clean samples after 95 epochs:
> > > Non-regularized model: 0.922
> > > With Adversarial Training: 0.865
> > > With d.d. Spectral Norm Regularization: 0.868
> > >
> > > Accuracy on adversarial samples after 95 epochs:
> > > Non-regularized model: 0.310
> > > With Adversarial Training: 0.625
> > > With d.d. Spectral Norm Regularization: 0.633
> > >
> > > We would like to stress again that the purpose of this is to provide evidence for the reasonable assumption that our experimental findings do not change qualitatively when moving to this much more complex architecture, not to attempt to outcompete any other training method. We hope this is satisfactory. We are happy to include the final results in the camera-ready version of the paper.
> > > Please also see our response to your response below.

---

### Decision · Program_Chairs · 2019-12-19

**Decision:**

Reject

**Comment:**

This paper shows an theoretical equivalence between the L2 PGD adversarial training and operator norm regularization. It gives an interesting observation and support it from both theoretical arguments and practical experiments. There has been a significant discussion between the reviewers and authors. Although the authors made efforts in rebuttal, it still leaves many places to improve and clarify, especially in improving the mathematical rigor of the  proof and experiments using state-of-the-art networks.